# Evaluation of Peak Inspiratory Flow Rate in Hospitalized Palliative Care Patients with COPD

**DOI:** 10.3390/pharmacy11040113

**Published:** 2023-07-06

**Authors:** Joshua Borris, Heather Cook, Sulgi Chae, Kathryn A. Walker, Mary Lynn McPherson

**Affiliations:** 1Department of Practice, Sciences, and Heath Outcomes Research, School of Pharmacy, University of Maryland, Baltimore, MD 21201, USA; kathryn.a.walker@medstar.net (K.A.W.); mmcphers@rx.umaryland.edu (M.L.M.); 2MedStar Franklin Square Medical Center, Baltimore, MD 21237, USA; heather.k.cook@medstar.net; 3Department of Psychiatry, Kaiser Permanente, Santa Clara, CA 95051, USA; sulgichae2@gmail.com; 4MedStar Health, Columbia, MD 21044, USA

**Keywords:** dry powder inhaler, DPI, PIFR, peak inspiratory flow rate, COPD

## Abstract

Dry powder inhalers are an effective yet costly COPD medication-delivery device. Patients must possess a minimum peak inspiratory flow rate (PIFR) for inhaled medication to be properly deposited into the lungs. Hospitalized palliative-care patients with diminished lung function due to advanced COPD may not possess the minimum PIFR (30 L/min) for adequate drug delivery. This study aims to quantify PIFR values for hospitalized palliative-care patients with advanced COPD to evaluate whether these patients meet the minimum PIFR requirements. Hospitalized patients ≥18 years old with a palliative-care consultation were eligible if they had a diagnosis of advanced COPD (GOLD C or D). Patients were excluded if they lacked decision-making capacity or had a positive COVID-19 test within the previous 90 days. Three PIFR values were recorded utilizing the In-Check^TM^ device, with the highest of the three PIFR attempts being utilized for statistical analysis. Eighteen patients were enrolled, and the mean of the highest PIFR readings was 72.5 L/min (±29 L/min). Post hoc analysis indicated 99.9% power when comparing the average best PIFR to the minimum PIFR (30 L/min) but only 51.4% power when compared to the optimal PIFR (60 L/min). This study found that palliative-care patients possess the minimum PIFR for DPI drug delivery.

## 1. Introduction

Chronic obstructive pulmonary disease (COPD), the third leading cause of death in the United States (U.S.), affects over 16 million Americans per year and presents a significant economic and healthcare burden [1,2,3]. Approximately 10% of palliative-care patients have chronic respiratory diseases as their primary palliative diagnosis [4]. Dry powder inhalers (DPIs) are a common medication-delivery device used for many respiratory conditions, and they can be more expensive than traditional metered-dose inhalers (MDI) or nebulized solutions [5]. DPIs require breath actuation for effective medication administration, so they can be challenging to use, with only up to approximately 50% of patients demonstrating proper inhaler technique [5]. DPIs are also can be expensive, as the cash price of a 1-month supply for Advair Diskus^®^ 250 mcg–50 mcg can be approximately four to six times the price of a 1-month supply of albuterol–ipratropium nebulizer ampules [6,7]. Therefore, to improve patient outcomes and decrease healthcare costs, proper utilization of these devices is essential.

Peak inspiratory flow rate (PIFR) is the maximum flow rate a patient can accomplish in one inspiration [8]. As DPIs require breath actuation for drug administration, an adequate PIFR is required for optimal drug delivery. Each DPI device has a specific internal resistance, which is owing to the required minimum and optimal PIFR thresholds (see Table 1) [8]. Most DPI devices require PIFR readings ≥60 L/min to receive optimal benefit and minimum PIFR readings of ≥30 L/min to receive any benefit.

PIFR has been shown to directly impact drug administration for DPIs in a proportional manner, with higher PIFR values leading to greater drug deposition, which may translate to better health outcomes [15,16]. Harb and colleagues found that suboptimal PIFR readings correlated with a greater incidence of GOLD D classification and greater symptom burden [17]. Suboptimal PIFR values are thought to occur in about 40–50% of patients with COPD, although wide ranges of true prevalence have been noted [17,18,19]. For patients with suboptimal PIFR, utilization of a more traditional metered-dose inhaler (MDI) with a spacer or a nebulized solution may provide greater drug deposition into the lungs and result in better health outcomes. In clinical practice, measurement of PIFR values is not always feasible; thus, inhaler selection must be made based upon readily available patient- and agent-related variables.

Palliative-care patients with advanced COPD are at high risk of not meeting the minimum PIFR threshold for DPI drug delivery given they commonly have multimorbidity, are frequently older, and may have greater disability [20]. Currently, there is a dearth of evidence evaluating palliative-care patients’ PIFR values to help guide clinicians in the selection of inhaler delivery devices. This study aims to quantify PIFR values for hospitalized palliative-care patients with advanced COPD to evaluate whether these patients meet the minimum PIFR requirements for satisfactory drug delivery.

## 2. Materials and Methods

This study collected data using convenience sampling from inpatient palliative-care consultations received between January 2020 to November 2021 across four urban, community teaching hospitals. Within this health system, patients with serious, life-limiting illnesses are eligible for inpatient palliative-care consultations. Patients were eligible if they were categorized as Global Initiative for Chronic Obstructive Lung Disease (GOLD) class C or D. Patients in class C or D had at least one COPD exacerbation leading to hospitalization within the past year, which is associated with an increased risk of mortality [21]. The other determining factor for GOLD classification is the symptom burden, assessed by the COPD Assessment Test (CAT) or Modified British Medical Research Council (mMRC) questionnaire [22]. The original intent of this study was to include both palliative-care and hospice patients as two separate groups, but hospice patients were not included due to COVID-19 restrictions. Patients were excluded from the study if they lacked decision-making capacity to consent or received a positive test for COVID-19 in the past 90 days.

The authors (J.B., H.C., and S.C.) recruited eligible, consenting patients and utilized the In-Check^TM^ device to measure patient PIFR readings on three attempts. This device utilizes the patient’s forceful inhalation to measure PIFR values between 30–370 L/min with an accuracy of ±10% or ±10 L/min (whichever is greater) [23]. The In-Check^TM^ device only contains one resistance setting, and therefore, the same internal resistance was utilized for all patients. The research investigator provided a brief instruction and tutorial to the patient prior to the first attempt. If any error was noticed by the investigator upon any attempt, re-education was provided, and that attempt was voided. All three PIFR values were recorded; however, only the highest value of the three PIFR attempts was recorded for the primary statistical analysis. To reduce variability in lung-function-based external factors, the timing of assessments was limited to the following criteria: at least two hours after waking, three hours after any nebulizer administration or “just before the next dose” of a bronchodilator, and approximately 48 h prior to discharge to allow baseline lung function to return to pre-admission capacity. If the highest PIFR value was detected to be <30 L/min, the primary team was notified to consider a different inhaler delivery device if the patient was receiving a DPI. In addition to the PIFR values, demographic information was also collected, including patient age and sex, inhalers prescribed prior to admission, the number of hospitalizations within the preceding 12 months (related and unrelated to COPD exacerbations), mMRC score (if documented), GOLD classification (if documented), and use of inpatient oral or intravenous (IV) steroids.

To achieve 80% power with a type-I error rate of 5%, it was determined that 30 patients per group would be required to be recruited to detect a difference of 10% between PIFR rate measurement and minimum PIFR rate (30 L/min), assuming a variability of 20% in PIFR rate measurement. Descriptive statistics were used to assess mean and median PIFR for all patients as well as to characterize baseline demographic and clinical information. A one-sample *t*-test was utilized to detect a statistically significant difference between minimum PIFR and mean PIFR. A bivariate analysis utilizing the Mann–Whitney U-test and Spearman correlation coefficients were utilized to measure any association between PIFR values and age, sex, and mMRC score. Post hoc analyses were utilized to determine statistical power due to not achieving the desired sample size (G*Power Version 3.1.9.4, Universität Kiel) [24].

This protocol was approved by the MedStar Health IRB on 7 November 2019 (IRB# STUDY00001382) and University of Maryland IRB on 17 October 2019 (IRB# HP-00087800), and informed consent was obtained for all patients.

## 3. Results

A total of 18 patients were included in the interim and final analysis. The study was stopped early due to low recruitment, the COVID-19 pandemic, and the inability to meet the primary outcome at the interim analysis. Baseline demographic information and clinical information related to COPD are displayed in Table 2. The majority of patients were >65 years old, female, and utilized a DPI prior to admission, and all were classified as GOLD group D. Five patients (28%) were not treated with either an oral or IV steroid during their hospitalization.

Of the patients studied, the mean PIFR was found to be 74.5 L/min (±29 L/min), and the median PIFR was 72.5 L/min (range 40–150 L/min; see Figure 1). While all patients had a highest PIFR of ≥30 L/min, two patients had at least one PIFR reading of ≤30 L/min. Six patients (33%) produced a highest value that did not meet the optimal threshold (≥60 L/min).

Maintaining a type-I error rate of 5%, comparing the mean PIFR to minimum PIFR (30 L/min) provided 99.9% power and an effect size of 1.5. When omitting the best PIFR values >100 L/min, an effect size of 2.1 with 100% power was achieved. When comparing the median PIFR to optimal PIFR (60 L/min), post hoc power was determined to be 51.4% with an effect size of 0.5.

PIFR values were significantly associated with sex (*p* = 0.01), as females had a mean best PIFR of 66 L/min (±30 L/min), while males had a mean best PIFR of 90 L/min (±19 L/min). When comparing the minimum and mean PIFR values for males and females separately, power remained high at 96.5% for females and 99.9% for males. There was no association between best PIFR and age (*p* = 0.7) or mMRC score (*p* = 0.2), although both showed very weak and weak negative correlations, respectively

## 4. Discussion

This is the first study to evaluate the PIFR readings of hospitalized palliative-care patients with a diagnosis of advanced COPD. Despite not reaching the recruitment goal due to early study termination, this study was able to demonstrate, with a very small probability of a false negative, that PIFR readings for these patients were significantly higher than the minimum PIFR requirement for DPI use. Unfortunately, the sample size was not large enough to determine conclusively if palliative-care patients with advanced COPD possess greater than the optimal PIFR indicated for DPI use.

There is conflicting evidence regarding the impact of PIFR on health outcomes. Both Sharma and colleagues as well as Samarghandi and colleagues found no negative impact of PIFR <60 L/min on healthcare outcomes, including 90-day COPD readmissions or all-cause readmissions up to 180 days [25,26]. In contrast, Loh and colleagues found that a PIFR <60 L/min was associated with higher rates of 90-day COPD readmissions and shorter time to readmissions due to COPD, although there was no difference in all-cause readmissions up to 90 days or COPD-related readmissions up to 30 days [27] Furthermore, Alqahtani and colleagues found that the average PIFR for patients with COPD readmissions at 30 days was 50 L/min compared to 80 L/min in patients not readmitted, implying that lower PIFR scores correlate to more frequent readmissions [28]. In addition to this, Chen and colleagues found that PIFR-guided inhaler therapy, which utilized PIFR values to determine optimal inhaler selection and provide education on proper inhalation technique, significantly reduced severe acute exacerbations [29]. While suboptimal PIFR may impact COPD readmission rates, there are many significant factors that contribute to COPD outcomes, including smoking, age, cardiovascular disease, and diabetes mellitus [22]. While not conclusive, some studies show that a suboptimal PIFR is correlated with female sex, advancing age, shorter height, decreased functional ability, and weaker grip strength [8,17,25,30,31]. This study also showed an association between female sex and lower PIFR values. Height, functional ability, frailty, and grip strength were not measured in this study and thus could not be assessed.

Studies have demonstrated that patients with suboptimal PIFR who are prescribed nebulizers have improved clinical outcomes, including lower COPD and all-cause readmission rates, compared to those who are prescribed DPIs [27]. Mahler and colleagues studied short-term benefits in patients with suboptimal PIFR and found that nebulized long-acting beta agonists produced greater lung function test results at 2 h when compared to DPI formulations [32]. As most nebulizers utilize relaxed spontaneous breathing and do not require coordinated forceful inspirations as required by DPIs, nebulizers may be a more favorable medication-delivery device for those with suboptimal PIFR [33]. MDIs may be considered as an alternative to DPIs, but MDIs require proper hand–breath coordination and also require an adequate PIFR for optimal drug delivery, although the addition of a spacer may help overcome these barriers [8,29]. Given the results in this study (i.e., 33% of patients with PIFR readings of <60 L/min), it may be advisable to consider nebulizers as a more favorable formulation in this population to ensure optimal drug delivery; however, more studies are needed to provide definitive clinical guidance.

In this study, hospitalized palliative-care patients with advanced COPD were able to meet the minimum PIFR requirements for sufficient DPI delivery. It is not clear whether this ability extends to patients under hospice care, who may have significantly decreased functional ability, advanced age, and weak grip strength. Therefore, lower PIFR values may be more likely in a hospice population and may warrant reconsideration of DPI use. Many hospice patients still receive DPIs despite nebulizer solutions being readily available as an alternative [34]. A key future study for this area would be quantifying hospice patients’ PIFR readings to understand if they would benefit from continued DPI utilization. Other future valuable studies would include palliative-care patients with a primary diagnosis of COPD along with other respiratory diseases as well as a study exploring whether PIFR values for palliative-care patients are associated with changes in hospitalization, morbidity, and mortality.

A major strength of this study includes being a prospective, multisite study. This study does have several limitations. Firstly, patients were included that met the criteria for GOLD class C or D regardless of if the primary palliative diagnosis was COPD or respiratory, potentially leading to selection bias. Secondly, the In-Check^TM^ device can only measure PIFR values between 30–370 L/min (with a standard error of 10% or 10 L/min), so the device could not detect exact values below 30 L/min [21]. Thirdly, not all PIFR measurements could occur immediately prior to the next bronchodilator administration due to standard MAR administration timing. This may have falsely elevated PIFR values if it was measured too close to the last long-acting bronchodilator administration. However, patients’ PIFR values were consistently measured at least 3 h after a nebulizer treatment and at least 2 h after waking. Additionally, patients’ handgrip strength and functional ability scores were not taken into consideration, which may have impacted the interpretation of the results. Lastly, as a result of early termination, this study had a small sample size, although post hoc power was determined to be very high. While this pilot study has multiple limitations, this is the first study that captures PIFR readings in the palliative-care population and may lay the foundation for future research and inform clinical practice.

## 5. Conclusions

This study found that PIFR values for hospitalized palliative-care patients with advanced COPD were greater than the minimum PIFR required for adequate drug delivery from a DPI. However, not all patients produced PIFR values that met the criteria for optimal DPI drug delivery. Further studies are needed to evaluate optimal DPI utilization in this population to support clinical decision making and determine factors correlated with suboptimal PIFR.

## Figures and Tables

**Figure 1 pharmacy-11-00113-f001:**
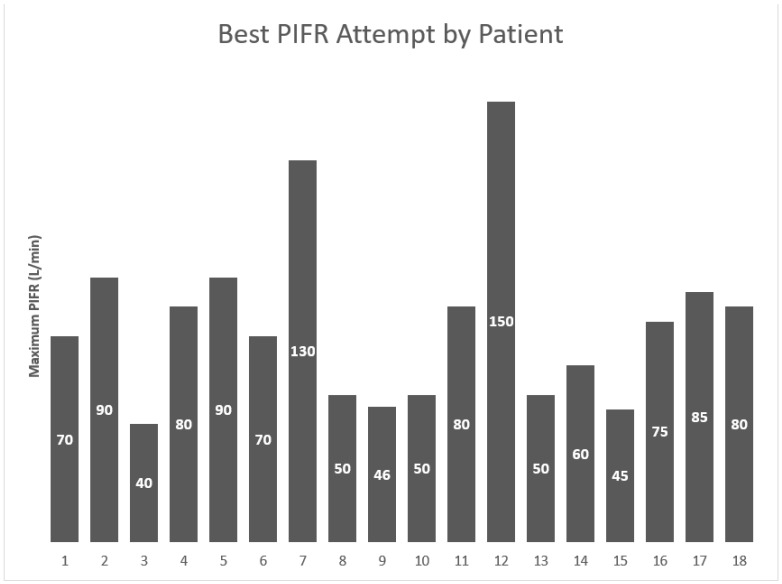
Best of three peak inspiratory flow rate (PIFR) attempts, stratified by patient.

**Table 1 pharmacy-11-00113-t001:** List of the optimal and minimum PIFR (peak inspiratory flow rate) values per the medication’s package insert [9,10,11,12,13,14].

Dry Powder Inhaler (DPI)	Optimal PIFR Recommended	Minimum PIFR Required
**Salmeterol Serevent Diskus^®^** (Brentford, UK)	60 L/min	30 L/min
**Formoterol Aerolizer^®^** (New York, NY, USA)	60 L/min	30 L/min
**Vilanterol and/or umeclidinium Ellipta^®^** (Brentford, UK)	60 L/min	30 L/min
**Aclidinium Genuair/PressAir^®^** (Wilmington, DE, USA)	63 L/min	30 L/min
**Tiotropium HandiHaler^®^** (Ridgefield, CT, USA)	39 L/min	20 L/min
**Indacaterol and/or glycopyrrolate Breezhaler/Neohaler^®^** (East Hanover, NJ, USA)	90 L/min	30 L/min

**Table 2 pharmacy-11-00113-t002:** Baseline demographic information and clinical criteria.

Demographic	*n* = 18 (%)
Median age, years (range)	69.5 (46–91)
Sex, female	12 (66%)
Inhaler prior to admission	
DPI MDI Nebulized	11 (61%) 6 (33%) 5 (28%)
Number of hospitalizations due to COPD exacerbations within past 12 months	
1 2 >3	5 (28%) 5 (28%) 8 (44%)
mMRC score	
0–1 2 3–4	0 (0%) 5 (28%) 13 (72%)
GOLD classification	
C D	0 (0%) 18 (100%)
Inpatient steroid use	
Oral IV	9 (50%) 10 (56%)

## Data Availability

The data presented in this study are available on request from the corresponding author. The data are not publicly available due to predetermined Institutional Review Board standards and procedures.

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
