# Peer review of "Evaluation of Peak Inspiratory Flow Rate in Hospitalized Palliative Care Patients with COPD"

_pharmacy, 2023, doi:10.3390/pharmacy11040113_

Round 1

Reviewer 1 Report

In-hospital PIFR assessment was a feasible tool to assess patient’s ability to use DRI. Although he idea of in-hospital PIFR assessment was not new but the population of palliative care has seldom assessed. There are some concerns as listed below.

1.      The definition of palliative care needs to be more specific. As outlined in page 2 column 76, patients receiving hospice care was excluded. Besides, the classification is not constant and would never be a good reason for palliative care. (Eur Respir J. 2021 Feb 11;57(2): 2002122).

2.      The benefit of PIFR-guided inhalation therapy has been noted and this should be discussed more as compared to current study (Front Pharmacol. 2021 Jun 29;12:704316).

3.      In column 80-85, the authors gave a brief introduction of how to measure PIFR. However, InCheck dial provided at least 4 resistances to be chosen from. Because PIFR would change in relation to resistance, the author should specify which resistance they choose when using InCheck. (Sci Rep. 2020 Apr 29;10(1):7271) Does every patient receive the same resistance? Is resistance applied (InCheck provides zero resistance)? This would significantly affect the PIFR value reported and the results.

4.      Many factors affect PIFR and female gender, height, number of previous exacerbations, and handgrip strength are well-documented factors. Please provide complete demographic data, including height, and handgrip strength in Table 2. Besides, COPD assessment test (CAT) score is also advised to be included in Table 2.

5.      Frailty was mentioned in column 177. It seemed that having palliative care was taken as equal to frailty in this study. However, there is a lack of subjective frailty assessment among the study group. Clinical frailty scale, short physical performance battery…etc. should be provided for readers to get a vivid picture of how frail the enrolled patients were. Or otherwise, the conclusion of minimum PIFR could be achieved by “frail” patients could not be made.

Reviewer 2 Report

This manuscript, even though the reduced number of patients, can be a start in the determination of the best inhalator device for patients with advanced COPD.

The different sections are correct, but something could be improved in the discussion

Why do the authors when comparing the suitability of DPI only refer to nebulizer? Do not they consider the possibility of MDI? Why?

Thinking in the discharge, for the patient at home it would be easier to use than a nebulizer.

It has been well stablished that nebulizer gives better results, but the what represents in the operability of the apparatus can be a problem if the patients has some mobility

Round 2

Reviewer 1 Report

The revision is appropriate.